# Evidence of the Autophagic Process during the Fish Immune Response of Skeletal Muscle Cells against *Piscirickettsia salmonis*

**DOI:** 10.3390/ani13050880

**Published:** 2023-02-28

**Authors:** Cristián A. Valenzuela, Marco Azúa, Claudio A. Álvarez, Paulina Schmitt, Nicolás Ojeda, Luis Mercado

**Affiliations:** 1Grupo de Marcadores Inmunológicos, Laboratorio de Genética e Inmunología Molecular, Instituto de Biología, Pontificia Universidad Católica de Valparaíso, Valparaíso 2374631, Chile; 2Laboratorio de Fisiología y Genética Marina, Centro de Estudios Avanzados de Zonas Áridas, Coquimbo 1781421, Chile; 3Facultad de Ciencias del Mar, Universidad Católica del Norte, Coquimbo 1781421, Chile; 4Laboratorio de Genética e Inmunología Molecular, Instituto de Biología, Pontificia Universidad Católica de Valparaíso, Valparaíso 2374631, Chile

**Keywords:** autophagy, skeletal muscle, fish, immune response, *Piscirickettsia salmonis*

## Abstract

**Simple Summary:**

In mammals, autophagy plays a fundamental role in the defense against intracellular pathogens; however, in fish, this noncanonical function has not been described. In this context, it was proposed to study whether autophagy was modulated/activated in muscle cells challenged with a bacterial pathogen. Muscle cell cultures were performed and challenged with *Piscirickettsia salmonis*, the main threat to the salmon industry. Genes associated with immune response and autophagy were evaluated. In addition, the protein content of the LC3-II-specific marker of the autophagic process was evaluated via Western blot. Additionally, genes associated with vesicular traffic and endocytosis were evaluated, finding that *P. salmonis* promotes these processes. The results show a concomitant modulation of the genes associated with the immune response, vesicular trafficking, and autophagy, suggesting an early intracellular response by the muscle cell against this bacterium. Due to the necessity of seeking and discovering new alternatives and strategies to fight intracellular pathogens in the salmon industry, a better understanding of how autophagy participates in immune system responses may lead to the development of new technologies that allow for the effective control of intracellular pathogens, improving animal welfare and contributing to the sustainability of the global fish industry.

**Abstract:**

Autophagy is a fundamental cellular process implicated in the health of the cell, acting as a cytoplasmatic quality control machinery by self-eating unfunctional organelles and protein aggregates. In mammals, autophagy can participate in the clearance of intracellular pathogens from the cell, and the activity of the toll-like receptors mediates its activation. However, in fish, the modulation of autophagy by these receptors in the muscle is unknown. This study describes and characterizes autophagic modulation during the immune response of fish muscle cells after a challenge with intracellular pathogen *Piscirickettsia salmonis*. For this, primary cultures of muscle cells were challenged with *P. salmonis,* and the expressions of immune markers *il-1β*, *tnfα, il-8*, *hepcidin*, *tlr3*, *tlr9*, *mhc-I* and *mhc-II* were analyzed through RT-qPCR. The expressions of several genes involved in autophagy (*becn1, atg9, atg5, atg12, lc3, gabarap* and *atg4*) were also evaluated with RT-qPCR to understand the autophagic modulation during an immune response. In addition, LC3-II protein content was measured via Western blot. The challenge of trout muscle cells with *P. salmonis* triggered a concomitant immune response to the activation of the autophagic process, suggesting a close relationship between these two processes.

## 1. Introduction

Autophagy is a highly conserved catabolic process essential for cellular homeostasis, degrading cytosolic macromolecules and damaged organelles to guarantee energy supply and nutrient availability for the cell. Furthermore, autophagy can modulate innate and adaptative immunity, contributing to the clearance of pathogens from the cell [1]. Under normal conditions, autophagy occurs at low levels in almost all cells, but is strongly induced under nutrient deprivation, amino acid starvation, cellular stress, cytokine exposure and pathogen infections [2,3]. Additionally, autophagy can act through a noncanonical pathway in antigen processing and presentation for MHC-I and MHC-II, restricting intracellular pathogens and regulating adaptive immunity [4]. At the molecular level, the autophagic pathway involves the concerted action of evolutionarily conserved ATG gene products involved in the initiation (*atg9*, *atg13* and *becn1*), autophagosome formation (*atg5*, *atg12* and *atg16*), the elongation of autophagic processes (*atg7* and *atg4*), and autophagosome closure (*gabarap* and *lc3*) [5]. Autophagy is triggered by activating a regulatory protein complex that recruits microtubule-associated light chain protein 3 (LC3) to the nascent autophagosome [6]. The conversion of LC3 into its active form, LC3-II, through conjugation to phosphatidylethanolamine (PE), allows for the transformation of the membrane into a closed double-membrane system [7]. Lastly, the autophagosome binds to the endosomal or lysosomal compartment, forming the autolysosome where organelles and macromolecules are degraded by a pool of proteolytic enzymes [8]. In mammals, autophagic activation is associated with immune signals and molecular pathways, involving toll-like receptor (TLR) activation through different pathogen-associated molecular patterns (PAMPs) [9]. For instance, autophagy activated by the endosomal TLR signaling pathway in macrophages plays a key role as a mechanism for Leishmania major infection resistance [10].

Mammalian skeletal muscle, a nonclassical immune tissue, possesses several membrane-bound TLRs with the ability to detect different PAMPs, triggering signaling pathways to activate autophagy [11]. These findings associate TLR signaling and autophagy, providing a potential molecular mechanism for the induction of autophagy in response to pathogen invasion. Therefore, the ability of TLR ligands to stimulate autophagy may be used to treat intracellular pathogens in mammals [12]. Moreover, several proinflammatory cytokines, such as IFN-γ, TNF-α, IL-1, IL-2, IL-6, and TGF-β induce autophagy, while anti-inflammatory cytokines such as IL-4, IL-10, and IL-13 display an inhibitory effect [13]. All these data suggest a close relation between the immune response and autophagy in mammalian skeletal muscle. However, this interaction is unknown in fish, and only a few reports related to autophagy exist. Nevertheless, due to the direct relation of autophagy and growth, a key parameter for fish-industry productivity, autophagic markers have been studied under different diets and nutritional trials. Results showed a direct association between fish’s energy requirement and the activation of autophagy [14]. For instance, the autophagic flux and the overexpression of autophagic genes *atg4*, *atg9*, *atg12*, *lc3*, *gabarap* and *becn1* was observed in vitro in epithelial gill cell line RT-gill-W1 of rainbow trout under nutrient restriction, and in the muscle of trout challenged with *Flavobacterium psychrophilum* [15,16]. Nevertheless, the modulation of autophagy in fish muscle cells during an immune-like response against an intracellular pathogen has not been elucidated.

This study aims to describe and characterize autophagic modulation through the immune response of fish muscle cells after a challenge with a critical bacterial pathogen for the salmon industry. Through the immune challenge of primary cultures of muscle cells with *Piscirickettsia salmonis*, a fastidious intracellular pathogen, we revealed the transcript upregulation and protein production of several immune markers and genes involved in autophagy. Results obtained in the present work give new insight into the role of autophagy in the immune response against bacterial pathogens in muscle cells.

## 2. Materials and Methods

### 2.1. Ethics Statement

The present research was approved by the bioethics committee of Pontificia Universidad Católica de Valparaíso (PUCV) BIOEPUCV-B 334-2020 and the Agencia Nacional de Investigación y Desarrollo (ANID) of the Chilean government, and adhered to all animal welfare procedures.

### 2.2. Fish Husbandry

In total, 24 rainbow trout (*Oncorhynchus mykiss*) with body weight of 10 ± 1.7 g and body length of 10 ± 1.5 cm were obtained from the Río Blanco Aquaculture Center (Valparaíso, Chile). These fish were kept in 3 tanks at a density of 8 fish per 25^-L^ tank, using 1 tank per trial. Fish were maintained at a temperature of 16 °C, fed daily with commercial feed, and provided with a photoperiod of 13 h of light and 11 h of darkness.

### 2.3. Primary Culture of Muscle Cells

Muscular cells were prepared from 6-8 g of the skeletal muscle of four rainbow trout as previously reported [17]. Briefly, the dorsal white muscle was collected and disintegrated in Dulbecco’s Modified Eagle’s Medium (DMEM) containing 9 mM NaHCO_3_, 20 mM HEPES, 10% horse serum at pH 7.4, 100 U/mL penicillin, and 100 µg/mL streptomycin (Sigma, St. Luis, MO, USA). After mechanical dissociation, the muscle was washed with DMEM and then digested with 0.2% collagenase solution in DMEM for 1 h at 18 °C. The suspension was centrifuged at 500 g for 5 min at 15 °C, and the resulting pellet was digested with a 0.1% trypsin solution in DMEM for 30 min at 18 °C. Trypsin was deactivated with complete medium (DMEM containing 10% fetal bovine serum, 100 U/mL penicillin, and 100 µg/mL streptomycin at pH 7.4), the suspension was filtered and centrifuged at 5000× *g* for 10 min at 18 °C, and 5 mL of DMEM was lastly added to the resulting pellet. The cell suspension was collected to count cells and evaluate viability with Trypan Blue 0.2%.

Cells were seeded at a density of 5 × 10^5^ per well in plates previously treated with poly-L-lysine (2 μg/cm^2^) and laminin (20 μg/mL). The cells were incubated at 18 °C for 14 days, 7 days in a proliferation medium containing DMEM, 9 mM NaHCO_3_, 20 mM HEPES, 10% fetal bovine serum, 100 U/mL penicillin, and 100 µg/mL streptomycin at pH 7.4, and 7 days in a differentiation medium composed of DMEM (9 mM NaHCO_3_, 20 mM HEPES, 100 U/mL penicillin, and 10 mg/mL streptomycin). This procedure was repeated three times (n = 3) per experimental trial described in the next sections.

### 2.4. Piscirickettsia salmonis Culture and Infection Protocol

A *P. salmonis* field isolate, Psal-104a, was obtained from the Chilean National Piscirickettsia salmonis Strain Collection at Pontificia Universidad Católica de Valparaíso (PUCV). The bacteria were cultured in basal medium (BM) broth composed of yeast extract (Merck, St. Louis, USA) 2.0 g L^−1^, peptone from meat (peptic digested, Merck) 2.0 g L^−1^, (NH_4_)2SO_4_ 1.32 g L^−1^, MgSO_4_·7H_2_O 0.1 g L^−1^, K_2_HPO_4_ 6.3 g L^−1^, NaCl 9.0 g L^−1^, CaCl_2_·2H_2_O 0.08 g L^−1^, FeSO_4_·7H_2_O 0.02 g L^−1^, and glucose 5 g L^−1^ at 18 °C at 100 rpm.

Exponentially growing *P. salmonis* (O.D. 600 = 0.3) was recovered via centrifugation at 5000× *g*, washed three times with saline-buffered solution (SBS: 0.15 M NaCl, 7.3 mM KH_2_PO_4_, 11.5 mM K_2_HPO_4_, pH 6.0) supplemented with 4% lactose, and resuspended in DMEM containing 9 mM NaHCO_3_, 20 mM HEPES, at pH 7.4. Bacterial cells were counted using a Petroff–Hausser chamber. Muscular cells were infected at a multiplicity of infection (MOI) of 10 for 4, 6, 8, and, 24 h. After the experimental times, the pathogen was removed and then sampled. As a positive control for autophagic activation, muscle cells were treated with rapamycin (50 nm), and DMEM was used as a vehicle for control, sampled at the beginning of the experiment (0 hpi).

### 2.5. Quantitative Real Time PCR (qPCR)

The total RNA from the primary culture of skeletal muscle was extracted according to the manufacturer’s guidelines for E.Z.N.A total RNA Kit I (Omega Bio-tek, Norcross, GA, USA), and quantified with NanoDrop LITE spectrophotometer (Thermo Scientific, Rockford, IL, USA). Residual genomic DNA was removed, and 1 µg of RNA was used for cDNA synthesis via RevertAid First strain cDNA Synthesis Kit (Thermo Scientific, Rockford, IL, USA) according to the manufacturer’s protocol. Real-time quantitative PCR (qPCR) assays were performed in a AriaMx Real-Time PCR System (Agilent Technologies, Santa Clara, CA, USA) with 15 µL volume reactions, of which 1 µL was cDNA (a 2-fold dilution), 0.2 μl of each primer (250 nM), and 8.8 µL of Takyon master mix (Eurogentec, Seraing, Belgium). The primers used in this study were designed and validated in our laboratory, and the sequence and efficiency of each primer are listed in Appendix A. Thermal cycling conditions were as follows: initial activation 95 °C for 10 min, 40 cycles of 30 s denaturation at 95 °C, 30 s annealing at 60 °C, and 30 s elongation at 72 °C, and each sample was loaded in duplicate. Relative expression analysis was conducted using geNorm software (https://genorm.cmgg.be/ (accessed on 30 March 2021)), and the results are expressed as fold changes using *EF-1α* and *b-actin* as housekeeping reference genes.

### 2.6. Western Blot

Total protein was extracted from primary culture cells in 150 µL Pierce^tm^ RIPA buffer (Thermo) supplemented with 8 mM EDTA, PMSF 1 mM, and a protease inhibitor cocktail (Calbiochem, San Diego, CA, USA), centrifuged at 12,000× *g*, and solubilized at 4 °C. Protein concentration was determined using a Pierce BCA Protein Assay Kit (Thermo Scientific, Hanover Park, IL, USA). The total protein (10 µg) was loaded into each line, separated with 18% SDS-PAGE, transferred to a nitrocellulose membrane, and blocked for 1 h at 37 °C in Tris-buffered saline (TBS-Tween) with 3% BSA. Primary antibody incubations (LC3A/B (D3U4C) XP^®^ Rabbit mAb #12741, Actine (INVITROGEN), and ant-Rab7a, diluted 1:1000, 1:2000 and 1:500, respectively) were performed overnight at 4 °C. The anti-Rab7a polyclonal antibody was developed in our lab. Briefly, a recombinant Rab7a was used for specific antibody production (Appendix A). CF-1 mice (5 weeks old) were subcutaneously injected at 1, 14, and 28 days with 30 μg of peptide diluted 1:1 in FIS, as a T helper cell activator, and in 1:1 Freund’s adjuvant (Thermo Scientific). The antiserum was collected on Day 42 and centrifuged at 700 g for 10 min, and the supernatant was stored at −20 °C. Antibody validation was determined via Western blot (Appendix A). Membranes were washed with 1X TBS and incubated for 1 h at room temperature with the appropriate secondary antibody. After washing, the membranes were visualized using commercial kit WESTAR SUPERNOVA from Cyanagen. Then, the membrane was visualized with the ChemiDoc Imaging Systems Bio-Rad system (Bio-Rad Laboratories, Hercules, CA, USA). Digitized images were used for the densitometric analyses of the bands in Image J software(National Institutes of Health, Bethesda, Maryland, USA). Differences are expressed as fold changes in protein content. Western blots were performed for all individual samples per experimental treatment; however, only one representative blot film is shown in the figures (Appendix A).

### 2.7. Immunofluorescence

Adherent cells were washed with 1X PBS and fixed with 4% paraformaldehyde (PFA) (in 1x PBS) for 10 min and permeabilized/blocked with 3% BSA with 0.3% Triton in 1X TBS for 30 min, and the quenching solution (50 mM ammonium chloride) was added for 10 min. Then, the cells were incubated overnight with the anti-LC3 antibody diluted at 1:100 (LC3A/B (D3U4C) XP^®^ Rabbit mAb #12741 in 1% BSA at room temperature. Samples were washed with 1X PBS and 0.02% TBST, and incubated with a commercial secondary antibody (1:750) (Invitrogen, Thermo Scientific) for 90 min at room temperature. Nuclear staining was performed with DAPI (Vector Laboratories) for conventional fluorescence microscopy, following the manufacturer’s instructions. For image capture, the samples were placed in a Leica CTR5000fluorescence microscope (Leica Microsystems, Wetzlar, Germany). Western blots were analyzed using the LC3-II/actin and Rab7a/actin ratios.

### 2.8. Statistical Analysis

All generated data were evaluated with one-way ANOVA and Student’s t test. Data are expressed as the mean (n = 3 per treatment) ± standard error of the mean (SEM), and *p* < 0.05 was considered statistically significant. All analyses were performed using general linear models on Graph Prism 7.0 software (GraphPad Company, Boston, MA., USA).

## 3. Results

### 3.1. Immune-like Response of Skeletal Muscle Cells after P. salmonis Challenge

To understand and characterize the response of muscle cells against bacterial pathogens, rainbow-trout skeletal muscle cells were challenged with *P. salmonis* for 6, 8, and 24 h. Next, immune molecular markers *tlr3*, *tlr9*, *il-1β*, *tnfα*, *il-8, hepcidin, mhc-I* and *mhc-II* were measured with RT-qPCR, and results showed a concomitant overexpression of toll-like receptors 3 and 9 after 6 h of challenge (Figure 1A). Proinflammatory cytokines *il-1β*, *tnfα* and *il-8* were modulated compared to the noninduced controls, where *tnfα* and *il-8* were upregulated significantly after 6 and 8 h after the challenge, respectively (Figure 1B,C). Antimicrobial peptide hepcidin increased after 6, 8, and 24 h after the challenge, but no statistical significance was observed (Figure 1C). No changes were observed in the mRNA levels of *mhc-I* and *mhc-II* during the entire trial (Figure 1D).

### 3.2. Autophagic Modulation in Muscle Cells after P. salmonis Challenge

To describe the modulation of autophagy in muscle cells after bacterial challenge, autophagic markers were analyzed via RT-qPCR, Western blot, and immunofluorescence. Rapamycin was used as the positive control of autophagic activation. Autophagic genes associated with the induction of this process were upregulated, particularly becn1, of which the mRNA level was significantly increased after 6 and 24 h of the challenge (Figure 2A). In the case of autophagosome formation genes, the overexpression of atg5 was observed after 6 h of treatment, while the transcript level of atg12 was downregulated after 8 h (Figure 2B). The mRNA levels of atg4, an elongation of an autophagosome molecular marker, was downregulated after 8 h of challenge (Figure 2C). Lastly, the transcript level of lc3 was significantly downregulated after 8 h of challenge (Figure 2D).

At the protein level, protein autophagic marker LC3-I/LC3-II was evaluated via immunofluorescence and Western blot. Agreeing with the gene expression of autophagic induction gene analysis, the protein content of LC3 showed an increase in fluorescence signal in the muscular cells challenged with *P. salmonis* compared with the control group at 6 h after challenge (Figure 3A). An increase in the LC3-II protein content was observed through Western blot analysis after 6 h of stimuli (Figure 3B). In addition, the mRNA levels of small GTPase proteins rab5a and rab7a were upregulated at 6 h after challenge, while the transcript level of rab11a was increased after 24 h (Figure 3C). Lastly, the protein content of Rab7 showed an increasing trend through Western blot (Figure 3D).

## 4. Discussion

In mammals, the ability of muscles to respond and participate in the immune response against pathogens and in autoimmune events is well-characterized [18]. This ability is represented by the immunocompetent attributes of muscle cells, such as the expression of costimulatory molecules, antigen-presenting machinery, proinflammatory cytokines, and pathogen recognition receptors [19,20,21,22,23]. Moreover, the interaction between muscle cells and resident immune cells is an essential step for wound healing and damage regeneration in this tissue [24,25].

In fish, the skeletal muscle appears as an active immunological organ, where immune reactions occur after an in vivo challenge with bacterial pathogens [26,27]. However, the ability of a muscle cell to express classical immune-like molecules by itself is still under study, and only few reports address how this type of cells may respond against PAMPs and pathogens [28,29]. For example, rainbow-trout myotubes responded against *Piscirickettsia salmonis* through the upregulation of *tlr1*, *tlr22,* and *il-8* at 8 h after a challenge [17]. In the present study, the immune response against *P. salmonis* generated a concomitant overexpression of *tnfα*, *tlr3,* and *tlr9* after 6 h of challenge. Toll-like receptors 3 and 9 are associated with an intracellular/antiviral response where these receptors can detect double-strand RNA and CpG DNA, respectively [30]. Reports in mammals showed how these receptors can be involved in diminishing the growth of intracellular bacterium Salmonella typhimurium [31]. Salmon head kidney cells (SHK-1) respond to an infection with *P. salmonis* through the overexpression of *tlr1* and *tlr5s* [32]. Furthermore, *P. salmonis* could induce a flagellin-dependent TLR5 activation in Atlantic salmon, which resulted in the upregulation of proinflammatory genes [33]. The overexpression of intracellular receptors suggests that intracellular defense mechanisms could be activated in response to *P. salmonis* infections. Despite the overexpression of markers associated with an intracellular response, the RNA levels of molecular markers *DotA* and *DotB* of *P. salmonis* were not detected via RT-qPCR (data not shown), which could indicate that, even though the bacterium does not enter the cell, the intracellular defense mechanisms are stimulated. On the other hand, proinflammatory cytokine *tnfα* represents an early response of muscular cells against bacterial pathogens. Indeed, an increase in NF-κB and TNFα at 48 h after a challenge against *Vibrio anguillarum* was observed on fish muscle [34]. Additionally, at 8 h after a challenge, the mRNA level of *il-8* was upregulated. In higher vertebrates, TNFα induces the expression of IL-8 [35]. This kind of response may be associated with the need for muscle cells to recruit immune cells to the site of infection, since IL-8 is a chemoattractant for leukocytes [36]. These results contribute to existing data confirming that the muscle cell responds to a bacterial challenge by itself. However, communication with the immune resident’s cells is needed to deploy an effective and robust response.

In mammals, the participation of autophagy in the immune response is well-characterized, where autophagy-related gene (ATG) proteins mediate direct pathogen degradation, inflammation restriction, antigen presentation on MHC molecules and survival of memory lymphocyte populations [37]. This process has mainly been studied for its role in the internal quality control of the cell, and for its implication in metabolism and cancer. However, autophagy can participate through a noncanonical pathway to respond against other dangerous stimuli, such as pathogen infection [38]. In fish, the modulation of this process in a pathological context is unclear, and only a few reports show how autophagy is modulated by nutrient restriction and pharmacological treatments [14,39].

Transgenic zebrafish and zebrafish cell lines were used as research tools on the autophagic regulation process, highlighting its role on the protection against pathogen infection, development, and lipid degradation [14]. Autophagy was also linked to development and extracellular matrix formation in zebrafish [40]. Recently, the modulation of autophagic genes *atg4*, *becn1*, *atg7*, *lc3*, *atg12,* and *gabarap* has been evaluated in rainbow-trout liver and muscle under different functional treatments, fasting, and a posteriori *F. psychrophilum* infection, suggesting a possible role of this process in the resistance against pathogens [16]. In the present work, autophagic genes were up- and downregulated in a time-dependent and coordinated way. The upregulation of *becn1* was observed at 6 h after challenge; *becn1* is a protein that facilitates the de novo formation of the phagophore, and is critical for the autophagic initiation in embryogenesis and antiviral responses in fish [41,42]. The next step in autophagy is autophagosome formation, where molecules *atg5* and *atg12* are directly involved [43]. We observed the upregulation of *atg5* at 6 h after a challenge with the subsequent downregulation of *atg12* after 8 h. Both genes are associated with intracellular immune response against viruses in fish, where *atg5* plays crucial roles in viral replication via promoting autophagy in orange-spotted grouper [44], while *atg12* can induce both autophagy and the Type I IFN response in large yellow croaker [45]. These two molecules are regulated by RAB5, a small GTPase, which facilitates the early formation of the autophagosome [46]. Six hours after a challenge with *P. salmonis*, the mRNA levels of *rab5a* and *rab7* increased significantly, suggesting the activation of vesicle traffic in response to the bacteria. In mammals, Rab7a is associated with the endocytic pathway [47]; in fish, this molecule is related with an antiviral response [48], suggesting the activation of different intracellular mechanisms in response to *P. salmonis*. Lastly, the genes involved in the elongation and closure of the autophagosome were downregulated after 8 h after challenge, particularly *atg4* and *lc3*. Altogether, the present data suggest a tight relation between autophagy and intracellular innate immune response, and are consistent with the used intracellular bacterial pathogen and the evaluated immune markers in our study.

A crosstalk between the immune response and autophagy has been reported in mammals. For example, TLRs were directly involved with autophagic activation after TLR ligand stimulations with poly I:C and LPS, among other PAMPs [9]. In addition, the stimulation of TLR7 with different ligands activated autophagy, eliminating intracellular microorganisms even when the target pathogen had not been canonically associated with TLR7 signaling [12]. In this context, the recognized ability of TLR ligands to stimulate autophagy was proposed as a prophylactic alternative to treat intracellular pathogens [12]. In the present study, *tlr3*, *tlr9*, *tnfα* and autophagic genes *becn1* and *atg5* were upregulated at the same experimental time (6 h after challenge), suggesting a relation between the autophagy and immune response of muscle cells against *P. salmonis*. This relation was confirmed in mammals where TNFα and other cytokines could induce autophagy [13]. Lastly, to corroborate the activation of autophagy in rainbow-trout muscle cells, we assessed the protein level of LC3-II, a classical protein marker for autophagy. Finding higher protein content of LC3-II in the treated cells added to the presented overall results and the background information, suggesting that the muscle cell deploys an immunelike response where autophagy may play an important role for the clearance of the pathogen. Deeper functional analysis and characterization are required to corroborate the dialogue and regulation between the two essential cellular functions.

## 5. Conclusions

The skeletal muscle of fish is a fundamental energy source for the organism and a recurrent site of infection where immunelike responses against different bacterial pathogens occur. The present study described for the first time the modulation of the immune response and its possible connection with autophagy in rainbow-trout muscle cells. The challenge of trout muscle cells with *P. salmonis*, an intracellular pathogen, triggered an immune response concomitant to the activation of the autophagic process. Results suggest an early response by the muscle cells against the bacteria just after only 6 h of challenge, where both immune and autophagic markers were upregulated. Considering the obtained data and the available research information, two important questions arise regarding the possible implication of *P. salmonis* challenge in fish muscle: (1) is autophagy acting as an intracellular immune mechanism to fight this pathogen or is the bacterium using this mechanism to enter the cell and evade the immune response? These questions are relevant and represent a new line of research that must be answered. Due to the necessity of seeking and discovering new alternatives and strategies to fight intracellular pathogens in the salmon industry, a better understanding of how autophagy participates in immune system responses may lead to the development of new technologies that allow for the effective control of intracellular pathogens, improving animal welfare and contributing to the sustainability of the global fish industry.

## Figures and Tables

**Figure 1 animals-13-00880-f001:**
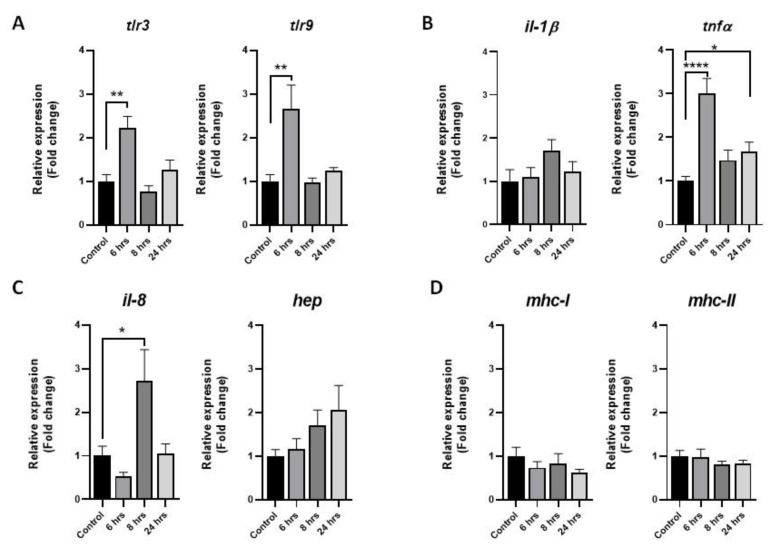
Expression of immune molecular markers in rainbow-trout muscle cells after *Piscirickettsia salmonis* challenge. Primary cultures of fish skeletal muscle were challenge with *P. salmonis* (MOI 10) for 6, 8, and 24 h. As control group, cells were treated with a vehicle (DMEM + 10% FBS). (**A**) Toll-like receptors 3 and 9. (**B**) Proinflammatory cytokines *il-1β* and *tnfα*. (**C**) *il-8* and the antimicrobial peptide *hepcidin*. (**D**) Major complex of histocompatibility I and II (*mhc-I* and *mhc-II*). qPCR analyses were normalized with the *ef-1α* and *β-actin* genes, and the results are presented as means ± SEMs of triplicates from three independent experiments, using fold change compared to the control values (n = 3). The difference in the transcript levels is denoted by asterisks (*) considering a *p*-value < 0.05 (*), *p*-value < 0,01 (**) and *p*-value < 0.001 (****) significant.

**Figure 2 animals-13-00880-f002:**
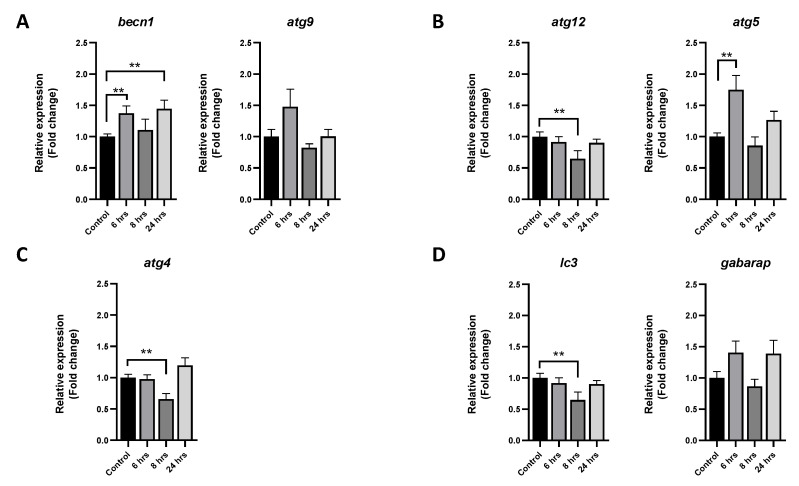
Effect of *P. salmonis* challenge on the expression of autophagic genes in rainbow-trout muscle cells. Primary cultures of fish skeletal muscle were infected with *P. salmonis* at MOI 10 for 6, 8, and 24 h. As the control, cells without bacteria were sampled at the beginning of the trial (time 0). (**A**) Autophagic genes *becn1* and *atg9* involved in the induction of this process. (**B**) Autophagosome formation genes *atg12* and *atg5*. (**C**) Gene *atg4,* involved in the elongation of the autophagosome. (**D**) Vesicle-completion-associated genes *gabarap* and *lc3*. The data are presented as means ± SEMs of triplicates from three independent experiments, using fold change compared to the values of the time 0 group (*n* = 3). qPCR analyses were normalized with the ef-1α and β-actin genes. The difference in transcript levels is denoted by asterisks (**) when the main effects were significant (*p* < 0.01).

**Figure 3 animals-13-00880-f003:**
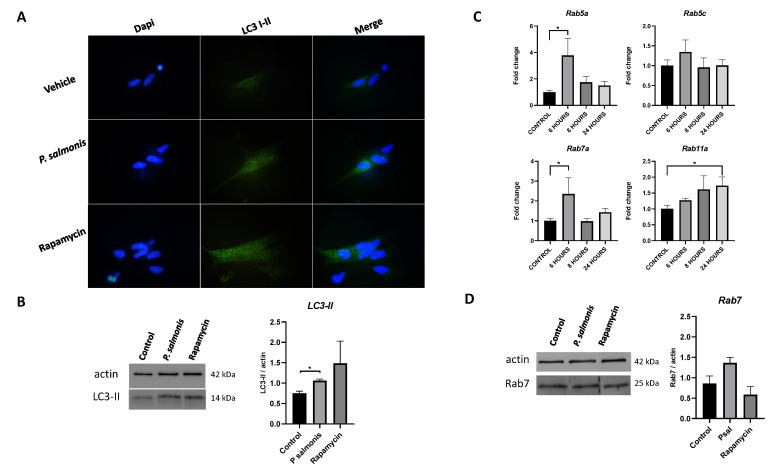
Effects of *P. salmonis* on the level of autophagy and Rab protein markers in muscle cells challenged. Fish skeletal muscle cells were cultured and challenge with *P. salmonis* at MOI 10 (resuspended in differentiation medium) for 6 h. (**A**) Immunofluorescence of LC3 (LC3-I and LC3-II). (**B**) Representative Western blot of LC3-II. (**C**) Rab small GTPases gene expression and, (**D**) Representative Western blot of Rab7a. The data are presented as the means ± SEMs of triplicates from three independent experiments, using fold change compared to the control values (untreated cells, *n* = 3). The measure of actin was used as load control for Western blots. qPCR analyses were normalized by the *ef-1α* and *β-actin* genes. The difference between the control group and treated groups is denoted by asterisks (*) when the main effects were significant (*p* < 0.05).

## Data Availability

Not applicable.

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
