# Peer review of "Evidence of the Autophagic Process during the Fish Immune Response of Skeletal Muscle Cells against Piscirickettsia salmonis"

_animals, 2023, doi:10.3390/ani13050880_

Round 1

Reviewer 1 Report

In addition to the maintenance of cellular homeostasis, autophagy also plays a critical role in eliminating pathogens that reside inside the cell. The zebrafish model has attracted attention for the modulation or manipulation of autophagy by a variety of bacterial pathogens. In the current study; using the intracellular pathogen Piscirickettsia salmonis, Valenzuela et al analyzed the expression of key genes /proteins involved in autophagy in trout muscle cells during infection.

In this manuscript, the methods used are standard for this type of study, the paper has been well written, and the data have been analyzed using appropriate statistics. There are a few suggestions listed below that can be used to further improve the manuscript.

MAJOR POINTS

Major point 1 : Line 89 – ‘effect of autophagy on fish skeletal muscle immune-like responses has not been elucidated’  - Liu et al, 2022 has studied the autophagy gene expression in rainbow trout muscle upon experimental infection with Flavobacterium

Liu, J.-T., Pham, P. H., & Lumsden, J. S. (2022). Autophagy modulation in rainbow trout Oncorhynchus mykiss L. and resistance to experimental infection with Flavobacterium psychrophilum. Journal of Fish Diseases, 45, 535– 545. https://doi.org/10.1111/jfd.13578

It would be great if the authors can cite this paper in the introduction and compare relevant results in the discussion

Major point 2  : Omission of p62 : It has been demonstrated that p62 (SQSTM1) plays an important role in TLR-mediated autophagy in innate immunity. P62 was not included in the analysis of the modulation of autophagy in rainbow trout muscle cells after Piscirickettsia infection. It is expected that the authors will comment on this critical omission.

Major point 3 immune evasion utilizing autophagy: A note on the role of Piscirickettsia salmonis is known to have a number of mechanisms by which it evades the immune system and is able to survive inside the cell. It would be interesting for the authors to comment on some of these factors of possible immune evasion utilizing autophagy in light of their findings

Minor comments

1. Italicize all gene names  ( when referring to mRNAs) and  also  binomial names of organisms

2. Line 123 -  Please seeded at a density of 5×105 per well   -   Use  proper superscript ie ‘ 105 cells per well

3. In the case of chemical formulae, use the proper subscript

4. The manuscript could be improved further by a few minor corrections in grammar and usage.

Author Response

Dear Reviewer, thanks for all the comments about our research article.

Reviewer 2 Report

In this investigation the authors provided evidence of autophagy process during immune response of skeletal muscle cells against piscririckettsia salmonis. The manuscript has potential for high impact and I have the following concerns that the authors may wish to consider

1.       I suggest removing the word “first” from the title

2.       Lines 91- replaced the word “trough” with through

3.       Lines 110- any reason for higher light period of 13 hours to dark period of 11 hours?

4.       Lines 146- can the authors mention the other hpi used in the investigation

5.       Throughout the manuscript 4 oC and 4oC is being used, sometimes the authors have it with space other times they don’t can the authors chose one pattern throughout the manuscript.

6.       Line 201- the authors mentioned that data were expressed as mean n=9 while in some of their studies they mentioned they used n=3 can they correct the statement?

7.       Line 208- several immune molecular markers can the authors list the immune marker they measured or are referring to?

8.       The figures bars that are not significantly different from each other do not have any sign indicating that they are similar. Can the authors add the signs to the bars?

Author Response

Dear Reviewer, thanks for the comments about our research article.

Round 2

Reviewer 2 Report

The authors have addressed all my concerns.